# EMERGENT COMPLEXITY VIA MULTI-AGENT COMPETITION

**Trapit Bansal**[*]  **Jakub Pachocki**  **Szymon Sidor**  **Ilya Sutskever**  **Igor Mordatch**
UMass Amherst  OpenAI  OpenAI  OpenAI  OpenAI

## ABSTRACT

Reinforcement learning algorithms can train agents that solve problems in complex, interesting environments. Normally, the complexity of the trained agent is closely related to the complexity of the environment. This suggests that a highly capable agent requires a complex environment for training. In this paper, we point out that a competitive multi-agent environment trained with self-play can produce behaviors that are far more complex than the environment itself. We also point out that such environments come with a natural curriculum, because for any skill level, an environment full of agents of this level will have the right level of difficulty.

This work introduces several competitive multi-agent environments where agents compete in a 3D world with simulated physics. The trained agents learn a wide variety of complex and interesting skills, even though the environment themselves are relatively simple. The skills include behaviors such as running, blocking, ducking, tackling, fooling opponents, kicking, and defending using both arms and legs. A highlight of the learned behaviors can be found here: https://goo.gl/eR7fbX.

## 1 INTRODUCTION

Reinforcement Learning (RL) is exciting because good reinforcement learning algorithms exist (Mnih et al., 2015; Silver et al., 2016; Schulman et al., 2015a; Mnih et al., 2016; Schulman et al., 2015b; Lillicrap et al., 2015; Schulman et al., 2017), allowing us to train agents that accomplish a great variety of interesting tasks. We can train an agent to play Atari games from pixels (Mnih et al., 2015) or get humanoids to walk (Schulman et al., 2017). RL is exciting partly because it is easy to envision an RL algorithm producing a broadly competent agent when trained on an appropriate curriculum of environments.

In general, training an agent to perform a highly complex task requires a highly complex environment, and these can be difficult to create. However, there exists a class of environments where the behavior produced by the agents can be far more complex than the environments; this is the class of the competitive multi-agent environments trained with self-play. Such environments have two very attractive properties: (1) Even very simple competitive multi-agent environments can produce extremely complex behaviors. For example, the game of Go has very simple rules, but the strategies needed to win are extremely complex. This is because the complexity of these environments is produced by the competing agents that act in it. Thus, as the other agents become more competent, the environment effectively becomes more complex. (2) When trained with self-play, the competitive multi-agent environment provides the agents with a perfect curriculum. This happens because no matter how weak or strong an agent is, an environment populated with other agents of comparable strength provides the right challenge to the agent, facilitating maximally rapid learning and avoiding getting stuck.

Self-play in competitive multi-agent environments is not a new idea – it has already been explored in TD-gammon (Tesauro, 1995) and refined in AlphaGo (Silver et al., 2016) and Dota 2 (OpenAI). In both cases, the resulting behavior was far more complex than the environment itself, and the self-play approach provided the agents with a perfectly tuned curriculum for each task. In this paper, we investigate whether the idea of competitive multi-agent environments can yield fruit in

---

[*]Work done as an intern at OpenAI. Correspondence to `tbansal@cs.umass.edu`

other domains: specifically, in the domain of continuous control, where balance, dexterity, and manipulation are the key skills.

In more detail, we introduce several multi-agent tasks with competing goals in a 3D world with simulated physics, using the MuJoCo framework (Todorov et al., 2012), where the agents would need to learn highly developed motor skills in order to succeed in the competitive environment. We train the agents using a distributed implementation of a recent policy gradient algorithm, Proximal Policy Optimization (Schulman et al., 2017). By adding a simple exploration curriculum to aid exploration in the environment we find that agents learn a high level of dexterity in order to achieve their goals, in particular we find numerous emergent skills for which it may be difficult to engineer a reward. Specifically, the agents learned a wide variety of skills and behaviors that include running, blocking, ducking, tackling, fooling opponents, kicking, and defending using arms and legs. Highlight of the learned behaviors on the various tasks can be found here: https://goo.gl/eR7fbX

## 2 PRELIMINARIES

In this section, we review some background on policy gradient methods, Proximal Policy Optimization and related work in the multi-agent reinforcement learning domain.

**Notation:** We consider multi-agent Markov games (Littman, 1994). A Markov game for $N$ agents is a partially observable Markov decision process (MDP) defined by: a set of states $\mathcal{S}$ describing the state of the world and the possible joint configuration of all the agents, a set of observations $\mathcal{O}^1, \ldots, \mathcal{O}^N$ of each agent, a set of actions of each agent $\mathcal{A}^1, \ldots, \mathcal{A}^N$, a transition function $\mathcal{T} : \mathcal{S} \times \mathcal{A}^1 \cdots \mathcal{A}^N \to \mathcal{S}$ determining distribution over next states, and a reward for each agent $i$ which is a function of the state and the agent's action $r^i : \mathcal{S} \times \mathcal{A}^i \to \mathbb{R}$. Agents choose their actions according to a stochastic policy $\pi_{\theta^i} : \mathcal{O}^i \times \mathcal{A}^i \to [0, 1]$, where $\theta^i$ are the parameters of the policy. For continuous control problems considered here, $\pi_\theta$ is Gaussian where the mean and variance are deep neural networks with parameter $\theta$. Each agent $i$ aims to maximize its own total expected return $R^i = \sum_{t=0}^{T} \gamma^t r_t^i$, where $\gamma$ is a discount factor and $T$ is the time horizon

**Policy Gradient:** Policy gradient methods work by directly computing an estimate of the gradient of policy parameters in order to maximize the expected return using stochastic gradient descent. These methods are behind much of the recent success in using deep neural networks for control (Schulman et al., 2015b; Heess et al., 2017; Lillicrap et al., 2015; Silver et al., 2016). Such methods are also attractive because they don't require an explicit model of the world. There are several different expressions for the policy gradient estimator which have the form $g := \mathbb{E}[A_t \nabla_\theta \log \pi_\theta]$. Different choices of $A_t$ lead to different algorithms, for example taking the sample return of a trajectory $A_t = \sum_t r_t$ leads to the REINFORCE algorithm (Williams, 1992). However, such algorithms suffer from high variance in the gradient estimates and it's typical to use a baseline, such as a value function baseline, to ameliorate the high variance. Generalized advantage estimation (Schulman et al., 2015b) takes this approach of using a learned value function to reduce variance at the cost of some bias and using an exponentially weighted estimator of the advantage function.

**Proximal Policy Optimization (PPO):** Achieving good results with policy gradient algorithms requires carefully tuning the step-size (Schulman et al., 2015a). Moreover, most policy gradient methods perform one gradient update per sampled trajectory and have high sample complexity. Recently, Schulman et al. (2017) proposed the PPO algorithm which addresses both these problems. This uses a surrogate objective which is maximized while penalizing large changes to the policy. Let $l_t(\theta) = \frac{\pi_\theta(a_t|s_t)}{\pi_{\theta_{old}}(a_t|s_t)}$ denote the likelihood ratio. Then PPO optimizes the objective:

$$L = \mathbb{E}\left[\min(l_t(\theta)\hat{A}_t, \text{clip}(l_t(\theta), 1 - \epsilon, 1 + \epsilon)\hat{A}_t)\right], \text{ where } \hat{A}_t \text{ is the generalized advantage estimate}$$

and $\text{clip}(l_t(\theta), 1 - \epsilon, 1 + \epsilon)$ clips $l_t(\theta)$ in the interval $[1 - \epsilon, 1 + \epsilon]$. The algorithm alternates between sampling multiple trajectories from the policy and performing several epochs of SGD on the sampled dataset to optimize this surrogate objective. Since the state value function is also simultaneously approximated, the error for the value function approximation is also added to the surrogate objective to compute the complete objective function (Schulman et al., 2017).

**Related Work:** Tan (1993) explored the multi-agent setting with independently learning agents using Q-learning, in particular exploring advantages of cooperative agents over independent agents in a 2D grid world. This was further explored by Matignon et al. (2012) again in the cooperative

setting. A lot of the work on multi-agent RL is focused on cooperative settings, see Busoniu et al. (2008) for a review of multi-agent RL and Panait & Luke (2005) for a review focused on cooperative settings. Stanley & Miikkulainen (2004) trained agents in a competitive 2D world, using evolutionary strategies to evolve both weights and structure of policies with competition as a fitness measure. Tampuu et al. (2017) studied the application of deep Q-learning to train Pong agents with competitive and collaborative rewarding schemes. He et al. (2016) used deep Q-learning to model competitive games where only one agent is learning and the Q network implicitly models the opponent. Silver et al. (2016) used self-play with deep reinforcement learning techniques to master the game of Go. Sukhbaatar et al. (2017) introduced a self-play method for generating an automatic training curriculum in single-agent environments. From a game-theoretic perspective, Heinrich & Silver (2016) studied fictitious self-play for achieving approximate Nash equilibrium in zero-sum games like Poker. Recently, Foerster et al. (2017a) introduced an algorithm which explicitly accounts for the fact that the opponent is also learning and showed that it can achieve cooperation in iterated prisoner's dilemma, however the algorithm requires access to the opponent's parameters. Recently, Lowe et al. (2017) and Foerster et al. (2017b) proposed methods for centralized learning in multi-agent domains, where the idea is to use an actor-critic method with a central critic which can observe the joint state and actions of all agents in order to reduce variance, evaluating on 2D games and StarCraft. In this work, we do not rely on centralized training and address the variance problem by using very large batchsize through a distributed implementation of the PPO algorithm. Moreover, we study fully competitive settings in a 3D world with simulated physics whereas prior applications have focused on toy 2D worlds or game-theoretic problems. Recent work on learning dexterous locomotion skills in 3D environments by adding complexity in the agent's environment (Heess et al., 2017) is also related. However, whereas Heess et al. (2017) learn complex behaviours by engineering complexity into the environment design and by engineering dense reward functions for these environments, the resultant complexity in our work is due to the presence of other learning agents in a simple environment. Our work is also related to early work in the graphics community (Sims, 1994) on evolving creature morphology in varying environments using genetic algorithm and work in animation (Wampler et al., 2010) for adversarial games. The competitive multi-agent learning framework is also related to generative adversarial networks (Goodfellow et al., 2014) and work on learning robust grasping policies through an adversary (Pinto et al., 2017).

## 3 COMPETITIVE ENVIRONMENTS

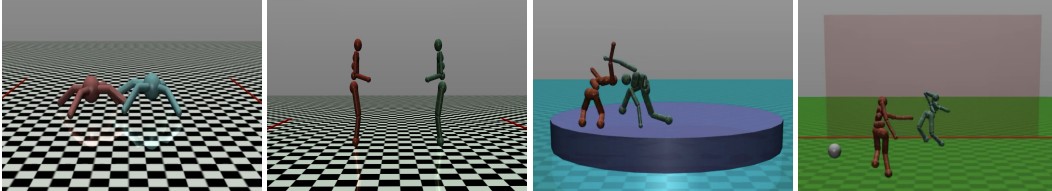

Figure 1: Illustrations of competitive environments we consider in our work: *Run to Goal*, *You Shall Not Pass*, *Sumo*, and *Kick and Defend*.

We introduce four competitive environments and experiment with two types of agents. In this paper we focus on two agent worlds, that is 1-vs-1 games, though these environments can be extended to include multiple agents for a mixed competitive and co-operative setup. We will now describe the four environments and the competitive rewards in each environment. Figure 1 shows a rendering of the environments. We consider two three-dimensional agent bodies: *ant* and *humanoid*. The ant is a quadrupedal body with 12 DoF and 8 actuated joints. Humanoid has 23 DoF and 17 actuated joints.

**Run to Goal:** The agents start by facing each other in a 3D world and they each have goals on the opposite side of the word (see Fig.1a). The agent that reaches its goal first wins. Reaching the goal before the opponent gives a reward of +1000 to the agent and -1000 to the opponent. If no agent reaches its goal then they both get -1000.

**You Shall Not Pass:** This is the same world as the previous task, but one agent (the blocker) now has the objective of blocking the other agent from reaching it's goal while not falling down. If the blocker is successful in preventing the opponent from reaching the goal and is standing at the end of

episode then it gets +1000 reward, if it is not standing then it gets 0 reward, and the opponent gets -1000 reward. If the opponent is successful in reaching it's goal then it gets +1000 reward and the blocker gets -1000 reward.

**Sumo:** The agents compete on a round arena (see Fig.1c) and the goal of each agent is to either knock the other agent to the ground or to push them out of the ring. The winner gets +1000 and the other agent gets -1000. If there is a draw then both agents get -1000.

**Kick and Defend:** This a standard penalty shootout (see Fig.1d). One agent has to kick a ball through the goal, which has a fixed width of 6 units, while the other agent defends. Successful kick or defend gives the agent +1000 reward and the opponent -1000 reward. The defender cannot go beyond the goal-keeping area which is a distance 3 units from the goal, doing so terminates the game with a penalty of -1000 for the defender. We give two additional rewards for defender: if defender is successful and it made contact with the ball then it gets additional +500 reward, and if the defender is successful and still standing at the end of the game then it gets another additional reward of +500. We found the latter two rewards to yield more realistic looking defending behaviors.

## 4 TRAINING COMPETITIVE AGENTS

In this section we describe the multi-agent training framework. We use a policy gradient algorithm, Proximal Policy Optimization (PPO) (Schulman et al., 2017), described previously. We adopt a decentralized training approach and use a distributed implementation of PPO for very large scale multi-agent training. This allows us to use really large batch-sizes during training ameliorating the variance problem to some extent while also aiding in exploration. Our distributed PPO implementation is similar to the implementation of Heess et al. (2017), where instead of the KL penalty we used the clipped objective as proposed in PPO (Schulman et al., 2017). We do multiple rollouts in parallel for each agent and have separate optimizers for each agent. We collect a large amount of rollouts from the parallel workers and for each agent optimize the objective with the collected batch on 4 GPUs. The approach is same as synchronous actor critic of Mnih et al. (2016). Instead of estimating a truncated generalized advantage estimate (GAE) from a small number of steps per rollout, as in Schulman et al. (2017); Heess et al. (2017), we estimate GAE from the full rollouts. This is important as the competition reward is a sparse reward given at the termination of the episode.

There are further challenges in applying distributed PPO to train multiple competitive agents. One is the problem of exploration with sparse reward and second is the choice of opponent during training which effects the stability of training. We now turn our attention to these issues.

### 4.1 EXPLORATION CURRICULUM

The success of agents in the competitive games requires the agents to occasionally solve the task (i.e. win the competition) by random actions. The probability of this happening in most games is minuscule as they require as a prerequisite some fundamental motor skills like the ability to walk. For example, the only way a kicker in the kick-and-defend task would achieve any positive reward is if it moves towards the ball and causes sufficient displacement to it so as to make it go past the goal boundaries which is also obstructed by a defender. This is a problem of training from sparse reward which is an active area of current research (Andrychowicz et al., 2017). To overcome this problem, we can use simple dense rewards at each step to allow the agents to learn basic motor skills initially. Such rewards have been previously researched for tasks like walking forward and standing up, see for e.g. Schulman et al. (2015b) and Duan et al. (2016). However, engineering such dense rewards for the competitive tasks is not straight forward. Moreover, such engineered rewards defeat the purpose of the competitive multi-agent training as we would like the agents to benefit from the natural curriculum arising from the multi-agent training. To overcome this chicken-and-egg problem, we instead propose to use a simple curriculum for training.

We use a dense reward at every step in the beginning phase of the training to allow agents to learn basic motor skills, like walking forward or being able to stand, which would increase the probability of random actions from the agent yielding a positive reward. We refer to this reward as the *exploration reward*. The exploration reward is gradually annealed to zero, in favor of the competition reward, to allow the agents to train for the majority of the training using the sparse competition reward. This is achieved using a linear annealing factor $\alpha$. So, at time-step $t$, if the exploration reward is $s_t$, the

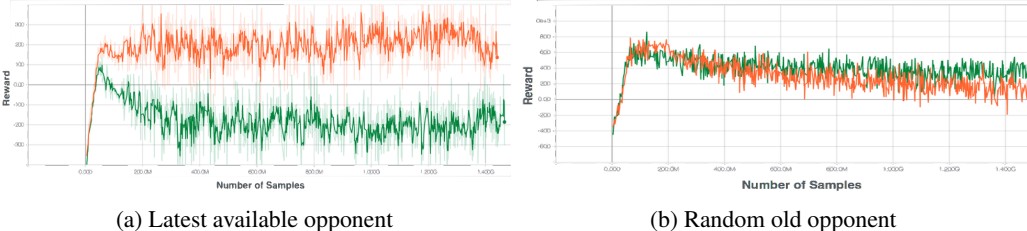

(a) Latest available opponent         (b) Random old opponent

Figure 2: Opponent Sampling: Training rewards for two opponent sampling strategies.

competition reward is $R$ and $T$ is the termination time-step, then the reward is:

$$r_t = \alpha_t s_t + (1 - \alpha_t)\mathbb{I}[t == T]R \tag{1}$$

This ameliorates the problem of exploration with the sparse reward, which is particularly tough in a 3D world with simulated physics and complex agents like humanoid, while still benefiting from training for the sparse competition reward for the majority of the training. During a typical training run, the agents would train on the dense reward for only about 10-15% of the training epochs. The dense rewards used are described in the Appendix A and are generally composed of the following terms: distance to goal, velocity in x-direction, control cost, impact cost, standing reward. These rewards are adopted from existing work and we did not tune weights on the various reward terms. In particular, it is important to note that there is no dense reward term for many of the complex emergent behaviors and we also show in the experiment section how the learned behaviors are affected if we do not anneal the dense reward to benefit from optimizing the sparse competition reward.

## 4.2 Opponent Sampling

In the competitive multi-agent framework, all agents are simultaneously training in opponent pairs. Thus, the skill of opponents encountered during training could have significant impact on the learning of the agents. We found that training agents against the most recent opponent leads to imbalance in training where one agent becomes more skilled than the other agent early in training and the other agent is unable to recover. Fig. 2a shows the rewards during training with this naive approach (for the "run to goal" task with ant). Instead, we found that training against random old versions of the opponent to work much better. Thus, during training, for each rollout for an agent we sample old parameters for the opponent. Fig. 2b shows the rewards for agents trained using this strategy. This leads to more stable training and more robust policies. We further analyze the effect of this opponent sampling in the experiments section. Note that for self-play this means that the policy at any time should be able to defeat random older versions of itself, thus ensuring continual learning.

## 5 Experiments

We train agents for the four competitive tasks using the training methods described previously. Our aim is to show that competitive multi-agent training provides a natural curriculum during learning which allows agents to learn complex behaviors. We provide additional training details and analyze various aspects of the competitive multi-agent training in this section. A highlight of the learned behaviors can be seen in the videos. Code for the environments as well as learned policy parameters for agents on all the environments are available: https://github.com/openai/multiagent-competition.

### 5.1 Experimental Details

**Policies and Value Functions:** We compare both MLP and LSTM for the policies and the value functions. MLP had 2 hidden layers with 128 units each. For LSTM networks, the input was first projected to a 128 dimensional embedding using a fully connected layer with ReLU activation which is then fed into a single-layer LSTM with 128 hidden state dimension and the output is projected to the action dimension using another fully connected layer. We used Gaussian policies with mean given by the output of the networks and a diagonal covariance matrix whose entries are also treated as trianable parameters. The policy outputs are clipped to lie within the control range. We used

MLP policy and value functions for the run-to-goal and you-shall-not-pass environments, and LSTM policy and value function for sumo and kick-and-defend. This is because earlier experiments did not yield good results with MLP policy on these tasks. For LSTM policy we used truncated BPTT with a truncation of 10 timesteps. The policy and the value functions have separate parameters. For the asymmetric games, you-shall-not-pass and kick-and-defend, we use separate policies for the two agents in a game.

**Observations:** For the Ant body we use all the joint angles of the agent, its velocity of all its joints, the contact forces acting on the body and the relative position and all the joint angles for the opponent. For the Humanoid body, in addition to the above we also give the centre-of-mass based inertia tensor, velocity vector and the actuator forces for the body. In addition to these, there are other environment specific observations. For the Sumo environment, we give the torso's orientation vector as the input, the radial distance from the edge of the ring of all the agents and the time remaining in the game. For kick-and-defend, we give the relative position of the ball from the agent, the relative distance of the ball from goal and the relative position of the ball from the two goal posts. Note that none of the agents observe the complete global state of the multi-agent world and only observe relevant sub-parts of the state vector to keep observations as close to real-world scenarios as possible.

**Algorithm Parameters:** We use Adam (Kingma & Ba, 2014) with learning rate 0.001. The clipping parameter in PPO $\epsilon = 0.2$, discounting factor $\gamma = 0.995$ and generalized advantage estimate parameter $\lambda = 0.95$. Each iteration, we collect 409600 samples from the parallel rollouts and perform multiple epochs of PPO training in mini-batches consisting of 5120 samples. For MLP policies we did 6 epochs of SGD per iteration and for LSTM policies we did 3 epochs. We don't use any entropy bonus. We found $l_2$ regularization of the policy and value network parameters to be useful. The co-efficient $\alpha_t$ in eq. 1 for the exploration reward is annealed to 0 in 500 iterations for all the environments except for kick-and-defend in which it is annealed in 1000 iterations.

## 5.2 Learned Behaviors

We observe numerous interesting learned behaviors demonstrated by the agents as a result of the complexity arising out of the competitive multi-agent training. Different random seeds often lead to somewhat different behaviors. Refer to the videos for highlights of the learned policies on all the tasks. On Run-to-Goal, we observe the quadruped Ants demonstrate behaviors like blocking, standing robustly, using legs to topple the opponent and running towards the goal. Humanoids try to avoid each other and run towards their goal really fast, occasionally they will bump into each other with force and try to recover from the impact. On You-Shall-Not-Pass, we observe the blocking humanoid learn to block by raising its hand while the other humanoid eventually learned to duck in order to cross. On Sumo, we observe multiple different strategies used by the Ant and Humanoid. Humanoids, for example, demonstrate a stable fighting stance and learned to knock the opponent using their heads. In a different run, we observe that one agent learned to charge towards the opponent whereas the opponent tried to fool it by stepping out of the opponents way at the edge of the ring. On kick-and-defend, we observe that the kicker learned a good kicking policy where it can go towards random ball positions, uses its feet to kick the ball high and tries to avoid the defender. We also see a fooling behavior in the kicker's motions where it moves left and right quickly once close to the ball to fool the defender. The defender learned to defend by moving in response to the motion of the kicker and using its hands and legs to obstruct the ball.

These movement strategies are not just useful in competition, for example the skills learned in the Sumo can transfer to other situations even without other agents. In one case, we took the agent trained on the multi-agent Sumo task and faced it with the task of standing while being perturbed by wind forces. The agent receives the zero vector for parts of the opponent observation. We found that the agent managed to stay upright despite never seeing the windy environment or observing wind forces. Please see Appendix B.1 for details of the experiment and quantitative results. Refer to the video for a demonstration.

## 5.3 Effect of Exploration Curriculum

In section 4.1 we introduced an exploration curriculum to help agents explore in a 3D world. One question that arises is the extent to which the outcome of learning is affected by this exploration reward and to explore the benefit of this exploration reward. As already argued, we found the

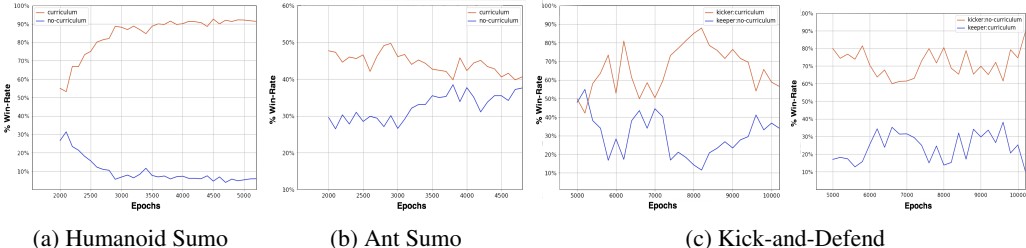

|     | (a) Humanoid Sumo | (b) Ant Sumo | (c) Kick-and-Defend |
|-----|-------------------|--------------|---------------------|

Figure 3: Effect for exploration curriculum: win-rate of agents trained by annealing the exploration reward against agents which constantly receive the dense exploration reward. The agents which optimized for the sparse competition reward benefit from the natural curriculum of multi-agent training and defeat the other agent by a margin.

| $\delta$ | 1.0 | 0.8 | 0.5 | 0.0 | $\mathbb{E}[\text{Win}]$ |
|----------|-----|-----|-----|-----|------|
| 1.0 | - | 0.26 | 0.13 | 0.37 | 0.25 |
| 0.8 | 0.46 | - | 0.22 | 0.52 | 0.40 |
| 0.5 | 0.59 | 0.58 | - | 0.73 | **0.63** |
| 0.0 | 0.55 | 0.36 | 0.16 | - | 0.35 |
| $\mathbb{E}[\text{Loss}]$ | 0.53 | 0.40 | **0.17** | 0.54 | - |

(a) Humanoid Sumo

| $\delta$ | 1.0 | 0.8 | 0.5 | 0.0 | $\mathbb{E}[\text{Win}]$ |
|----------|-----|-----|-----|-----|------|
| 1.0 | - | 0.37 | 0.35 | 0.29 | 0.34 |
| 0.8 | 0.36 | - | 0.38 | 0.33 | 0.36 |
| 0.5 | 0.36 | 0.39 | - | 0.33 | 0.36 |
| 0.0 | 0.51 | 0.49 | 0.49 | - | **0.50** |
| $\mathbb{E}[\text{Loss}]$ | 0.41 | 0.42 | 0.41 | **0.32** | - |

(b) Ant Sumo

Table 1: The effect of opponent sampling. $\mathbb{E}[\text{Loss}]$ and $\mathbb{E}[\text{Win}]$ are the expected loss and win-rates for agents trained with a particular $\delta$ as described in 5.4. For humanoid $\delta = 0.5$ gives highest win-rate and lowest loss, whereas for Ant $\delta = 0$ was best.

exploration reward to be crucial for learning as otherwise the agents are unable to explore the sparse competition reward. However, the learned behaviors are mostly a result of the natural curriculum arising out of the multi-agent competition and not due to the dense exploration reward. To see this, first note that we do not give any reward for many of the complex learned behaviours described previously. We further test this by not annealing the exploration reward and always having a dense reward which is a sum of the exploration reward and the competition reward. We take these agents trained without curriculum and pit them against agents trained with exploration curriculum. We plot the average win-rates over 800 games at various intervals during training in Fig. 3, for the sumo and kick-and-defend environments. For kick-and-defend there are two plots, one where kicker is trained with curriculum while keeper without it and vice versa. Observe that the agents trained with curriculum beat the non-curriculum agents by a margin. We found agents trained without curriculum exhibit either non-optimal behaviors for the competition or end up optimizing for a particular component of the dense reward. For example, for Sumo, the agents just learn to stand and move towards center of the arena, and for kick-and-defend, the defender optimizes for being able to stand up but doesn't learn to defend while the kicker learns a non-optimal strategy of carrying the ball with itself to the goal (rather than kicking) – a policy which is easily defeated by a defender trained with curriculum. Moreover, training without curriculum also takes more samples to learn. We also show these behaviours qualitatively in the videos. These results echo some recent findings (albeit in the single agent case), like Andrychowicz et al. (2017) who found that optimizing for the sparse reward yields better return than optimizing for hand crafted dense rewards. For the competitive multi-agent case, these results shed further light on the importance of the natural curriculum.

## 5.4 EFFECT OF OPPONENT SAMPLING

In section 4.2, we introduced the past opponent sampling method for training competitive agents simultaneously. This choice of opponent could be important as it affects the natural curriculum for the agents. We test different opponent sampling strategies by considering a threshold on the oldest opponent for each agent. That is, instead of uniform random over the entire history, we can consider sampling opponent from $\text{Uniform}(\delta v, v)$ where $v$ is the iteration number for the latest available parameters of the opponent and $\delta \in [0, 1]$ is a threshold. Thus, $\delta = 1.0$ corresponds to the latest available opponent and $\delta = 0.0$ corresponds to uniform sampling over the entire history. We train agents on the Sumo task via self-play, using a $\delta \in \{1.0, 0.8, 0.5, 0.0\}$ and pit the four agents against

each other to understand which sampling strategy leads to more robust policies. Since the agents have different skills and strengths at various points during training, we compute a Monte Carlo estimate of the expected win-rate for two agents that have seen the same number of samples taken at a random point during training. This is done by taking average of the win-rates of 30 agents at intervals of 100 iterations after a burn-in of 3000 iterations, where each win-rate is computed from an average over 800 episodes. Table 1a reports the results for Humanoid and Table 1b reports the results for Ant. First note that training against the latest opponent leads to worst performance, as argued earlier. Surprisingly, we found that uniform random ($\delta = 0.0$) over the entire history to have the highest win-rate for Ant and $\delta = 0.5$ to have the highest win-rate for Humanoid. This could be because Ant with random policy on a small arena is still a good opponent while a Humanoid with random policy is unable to stand and thus always looses in a few steps. The differences in these win-rates for different sampling strategies show that the choice of the opponent during sampling is important and care must be taken while designing training algorithms for such competitive environments.

## 5.5 LEARNING ROBUST POLICIES

Over-fitting to a particular dataset is often a problem in supervised learning. Similar problems can arise in reinforcement learning setups when there is no or little variation in the environment. We discovered two such problems in our competitive multi-agent training framework and we analyze and propose solutions to address these issues.

### 5.5.1 RANDOMIZATION IN WORLD

In order to learn robust policies which generalize better we can introduce randomness in the environment, for example the arena radius for the sumo environment can be randomized, the ball position for the kick-and-defend environment can be randomized, agent start positions can be randomized. However, we found that while randomization is crucial to learn policies which generalize better, it might hinder learning early on as there might be too many things for the agents to explore. Indeed, we observe that in kick-and-defend the agents are unable to learn to kick with a lot of randomization in both the ball and agent positions, whereas when trained with no randomization the learned policies are overfit to the particular position of the ball (see Fig. 4). Thus, in order to learn policies that generalize well, we introduce a simple curriculum in the randomization where we start with a small amount of randomization which is easier to solve and then gradually increase the randomization during training. We found this curriculum to work well for all the environments.

### 5.5.2 COMPETING AGAINST ENSEMBLE OF POLICIES

Another related problem that we observed is over-fitting to the behavior of the opponent when trained for very long. This results in policies which are good against particular types of opponents but do not generalize to other opponents (say opponents trained with a different random seed). This overfitting can also be observed in win-rates against opponent during training, where one would see oscillations as agents try to adapt to their particular opponent and changes in their strategies. To overcome this we propose learning multiple policies simultaneously. Thus, there is a pool of policies and in each rollout for a particular policy one of the other policies is selected at random as the opponent (in symmetric games, the same policy can also be an opponent). This is similar to multi-task learning (Caruana, 1998) where the same network is used to model multiple related tasks which allows sharing of statistical strength among tasks and reduces overfitting. In this case, the pool of all policies as opponents – current and throughout the history of training – creates a natural distribution over related tasks for multi-task learning. We found random policy initialization to provide enough diversity between agent policies, however techniques that explicitly encourage diversity (Liu & Wang, 2016) can potentially be incorporated in the future.

In order to test the robustness of training policies in an ensemble, we experiment on the Sumo environment with Ant and Humanoid bodies. We train a pool of three policies in an ensemble and take the policy with the highest average training reward in the last 500 iterations as the best ensemble policy. We also train three independent policies via self-play, that is just a single policy is trained in a run, and again take the policy with the highest average training reward in last 500 iterations as the best self-play policy. Then we pit the best ensemble policy against the best self-play policy and record average win-rates over 800 games. Fig. 5 shows the win-rates over training iterations

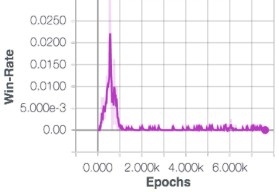 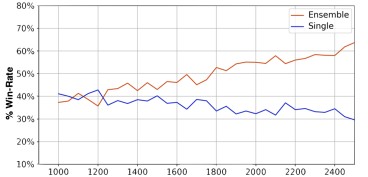 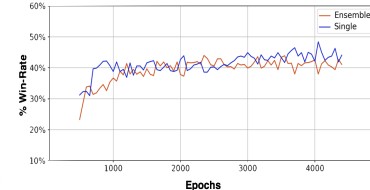

Figure 4: Win-rate of kicker vs iterations with full randomization

Figure 5: % Win-rate of agents trained in ensemble vs agents trained with just a single policy. Humanoid Sumo (left) and Ant Sumo (right).

(after 1000 iterations of training). We find that training in ensemble performs significantly better for the humanoid body, whereas for ant the performance is similar to training a single policy. Again we suspect this is because there is not enough variability in the behavior of ant across different runs. While training single policies might occasionally get stuck in a local minima and learn sub-optimal behaviors, we found that when training in an ensemble to be more robust to such minima. Qualitatively, we see more robust behavior of the humanoid trained in ensemble (see video).

## 6 CONCLUSION

We have presented several new competitive multi-agent 3D physically simulated environments. We demonstrate the development of highly complex skills in simple environments with simple rewards. In future work, it would be interesting to conduct larger scale experiments in more complex environments that encourage agents to both compete and cooperate with each other. Incorporation of additional skills, such as reasoning about other agents, potentially via techniques from Foerster et al. (2017a), may also be important in our setting.

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

## A  EXPLORATION REWARDS

We define the dense exploration rewards used for the tasks in this section. Our exploration reward terms are based on adapting the rewards defined previously for the training humanoids and quadrupeds to walk (Duan et al., 2016; Schulman et al., 2015b). We first review this locomotion reward and then define the task-specific dense rewards. These rewards take the form $r_t(s, a) = v_{fwd} + c_t(s, a) + C_{alive}$ where $v_{fwd}$ is the velocity in the forward direction, a bonus for standing $C_{alive}$ and costs for impact and action $c_t(s, a)$. We considered the following locomotion reward defined for Humanoid-v1 environment in OpenAI Gym package:

$$r_t^h(s, a) = v_{fwd} + c^h(s, a) + C_{alive} = v_{fwd} - 0.1||a||^2 - 5 \cdot 10^{-7}||F_{impact}||^2 + C_{alive} \quad (2)$$

where $F_{impact}$ is the contact force vector clipped to values between 1 and 1, and $C_{alive}$ is a bonus for the center of the body being at a certain height, defined as $C_{alive} = +5$ if $2.0 \geq z_{body} \geq 1.0$, else 0.

Similarly, the following is the reward for quadruped locomotion:

$$r_t^q(s, a) = v_{fwd} + c^q(s, a) = v_{fwd} - 0.5||a||^2 - 5 \cdot 10^{-4}||F_{impact}||^2 + C_{alive} \quad (3)$$

where $C_{alive} = +1$ if $1.0 \geq z_{body} \geq 0.2$, else 0.

In the following, superscript $h$ refers to humanoid agents and superscript $q$ refers to quadruped. We redefine $C_{alive}$ to be $+5$ if $z_{body} \geq 1.0$, else $-5$ for humanoid, and $C_{alive} = +1$ if $z_{body} \geq 0.28$, else $-1$ for quadruped.

**Run to Goal**  For humanoids, reward is $r^h(s, a) - |x - g|$ where $x - g$ is the $l_1$ distance of the agent from the goal $g$ along the $x$-axis. For ant, reward is similar $r^q(s, a) - |x - g|$.

**You Shall not Pass**  For the agent whose goal is to reach the other side, the reward is same as for run-to-goal. For the blocking agent, the reward for humanoid is $c^h(s, a) + C_{alive} + |x' - g|$ where $|x' - g|$ is the distance of opponent to the goal.

**Sumo**  For humanoids, reward is $c^h(s, a) + C_{alive} - (x^2 + y^2)^{0.5}$, where the last term is distance from the center of the ring. Similarly for ant: $c^q(s, a) + C_{alive} - (x^2 + y^2)^{0.5}$

**Kick and Defend:**  For kicker, reward is $r^h(s, a) - ||x - b|| - |b_x - g|$, where $b$ is the $(x, y)$ position of the ball, $b_x$ is the $x$-coordinate of the ball and $g$ is the $x$-coordinate of the goal-post. For defender, reward is $c^h(s, a) + C_{alive} + |b_x - g|$ where for $C_{alive}$ we only gave positive reward if the defender was in front of the goal area.

| | Force Magnitude | | | | |
|---|---|---|---|---|---|
| | 200 | 300 | 400 | 500 | 600 |
| Sumo Agent | $372 \pm 146$ | $327 \pm 150$ | $247 \pm 143$ | $181 \pm 114$ | $123 \pm 57$ |
| Walker Agent | $179 \pm 54$ | $139 \pm 42$ | $116 \pm 32$ | $103 \pm 23$ | $95 \pm 20$ |

Table 2: Average number of steps before agent falls. *Sumo Agent* refers to the agent trained in Sumo environment whereas *Walker Agent* refers to the agent trained to walk in a single agent environment.

## B  ADDITIONAL RESULTS

### B.1  TRANSFER RESULTS

We took the agent trained on the multi-agent Sumo task and faced it with the task of standing while being perturbed by wind forces. The agent receives a zero vector for parts of the observation space which correspond to the opponent. We calculate the number of steps before the agent falls down (i.e. when $z_{body} \leq 0.5$) or the agent is pushed out of the arena and report the average steps over 200 episodes. Episodes last a maximum of 500 time steps. In half the episodes the wind force is applied in a radially outwards direction and in the remaining half it is applied in the radially inwards direction. We allow 50 steps for the agent to stabilize and apply the force at intervals of 50 steps where in between the intervals the force magnitude is decayed at a constant rate:

$$F_t = \begin{cases} F & \text{if } t \equiv 0 \pmod{50} \\ 0.9 * F_{t-1} & \text{otherwise} \end{cases}$$

where $F \in \{200, 300, 400, 500, 600\}$.

We compare with a humanoid agent trained in a single agent environment for the task of walking. We used same LSTM policy architecture as used for the Sumo agent and trained the humanoid in the publicly available OpenAI Gym Humanoid-v1 environment using PPO. We then apply force on this agent using the same method as above where the direction of the force is in the direction the agent is walking in half the episodes and opposite to it in the remaining half. We record average number of steps to fall using the same condition as for the Sumo agent.

Table 2 shows the average number of steps over 200 episodes along with the standard deviation. We see that the humanoid trained in Sumo is more robust to adversarial forces and able to withstand large magnitude of force for many steps.

