# OpenReview forum: "Emergent Complexity via Multi-Agent Competition"
_ICLR.cc/2018/Conference — Accept (Poster)_

### Official Review · AnonReviewer2 · 2017-11-21
**This paper showing nice results lacks a serious scientific analysis and contains several issues**

**Rating:** 3
**Confidence:** 3

**Review:**

In this paper, the authors produced quite cool videos showing the acquisition of highly complex skills, and they are happy about it. If you read the conclusion, this is the only message they put forward, and to me this is not a scientific message.

A more classical summary is that the authors use PPO, a state-of-the-art deep RL method, in a context where two agents are trained to perform competitive games against each other. They reuse a very recent "dense reward" technique to bootstrap the agent skills, and then anneal it to zero so that the competitive rewards obtained from defeating the opponent takes the lead. They study the effect of this annealing process (considered as a curriculum) and of various strategies for sampling the opponents. The main outcome is the acquisition of a large variety of useful skills, just observed from videos of the competitions.

The main issue with this paper is the lack of scientific analysis of the results, together with many local issues in the presentation of these results.
Below, I talk directly to the authors.

---------------------------------

The related work subsection is just a list of works, it should explain how the proposed work position itself with respect to these works.


In Section 5.2, you are just describing "cool" behaviors observed from your videos.
Science is about producing quantitative results, analyzing them and discussing them.
I would be glad to read more science about these cool behaviors. Can you define a repertoire of such behaviors?
Determine how often they are discovered? Study how the are represented in the networks?
Anything beyond "look, that's great!" would make the paper better...

By the end of Section 5.2, you allude to transfer learning phenomena.
It would be nice to study these transfer effects in your results with a quantitative methodology.

Section 5.3 is more scientific, but it has serious issues.

In all subfigures in Figure 3, the performance of opponents should be symmetric around 50%. This is not the case for subfigures (b) and (c-1). Why?
Do they correspond to non-zero sum game? The x-label is "version". Don't you mean "number of epochs", or something like this? Why do the last 2 images
share the same caption?

I had a hard time understanding the message from Table 1. It really needs a line before the last row and a more explicative caption.

Still in 5.3, "These results echo"...: can you characterize this echo? What is the relationship to this other work?

Again, "These results shed further light": further with respect to what? Can you be more explicit about what we learn?

Also, I find that annealing a kind of reward with respect to another is a weak form of curriculum learning. This should be further discussed.

In Section 5.4, the idea of using many opponents from many stages of learning in not new.
If I'm correct, the same was done in evolutionary method to escape the "arms race" dead-end in prey-predator races quite a while ago  (see e.g. "Coevolving predator and prey robots: Do “arms races” arise in artificial evolution?" Nolfi and Floreano, 1998)

Section 5.5.1 would deserve a more quantitative presentation of the effect of randomization.
Actually, in Fig5: the axes are not labelled. I don't believe it shows a win-rate. So probably the caption (or the image) is wrong.

In Section 5.5.2, you "suspect this is because...".
The role of a scientific paper is to clearly establish results and explanation from solid quantitative analysis.

-------------------------------------------
More local comments:

Abstract:

"Normally, the complexity of the trained agent is closely related to the complexity of the environment." Here you could cite Herbert Simon (1962).

"In this paper, we point out that a competitive multi-agent environment trained with self-play can produce behaviors that are far more complex than the environment itself."
Well, for an agent, the other agent(s) are part of its environment, aren't they? So I don't like this perspective that the environment itself is "simple".

Intro:

"RL is exciting because good RL exists." I don't believe this is a strong argument. There are many good things that exist which are not exciting.

"In general, training an agent to perform a highly complex task requires a highly complex environment, and these can be difficult to create." Well, the standard perspective is the other way round: in general, you face a complex problem, then you need to design a complex agent to solve it, and this is difficult.

"This happens because no matter how weak or strong an agent is, an environment populated with other agents of comparable strength provides the right challenge to the agent, facilitating maximally rapid learning and avoiding getting stuck." This is not always true. The literature is full of examples where two-players competition end-up with oscillations between to solutions rather than ever-increasing skill performance. See the prey-predator literature pointed above.

"in the domain of continuous control, where balance, dexterity, and manipulation are the key skills." In robotics, dexterity, and manipulation usually refer to using the robot's hand(s), a capability which is not shown here.

In preliminaries, notation, what you describe corresponds to the framework of Dec-POMDPs, you should position yourself with respect to this framework (see e.g. Memory-Bounded Dynamic Programming for DEC-POMDPs. S Seuken, S Zilberstein)

In PPO description : Let l_t(\theta) ... denote the likelihood ratio: of what?

p5:
would train on the dense reward for about 10-15% of the trainig epochs. So how much is \alpha_t? How did you tune it? Was it hard?

p6:

you give to the agent the mass: does the mass change over time???

In observations: Are both agents given different observations? Could you specify which is given what?

In Algorithms parameters: why do you have to anneal longer for kick-and-defend? What is the underlying phenomenon?

In Section 5, the text mentions Fig5 before Fig4.

-------------------------------------------------
Typos:

p4:
research(Andrychowicz => missing space
straight forward => straightforward

p5:
agent like humanoid(s)
from exi(s)ting work

p6:
eq. 1 => Eq. (1) (you should use \eqref{})
In section 4.1 => In Section 4.1 (same p7 for Section 4.2)

"One question that arises is the extent to which the outcome of learning is affected by this exploration reward and to explore the benefit of this exploration reward. As already argued, we found the exploration reward to be crucial for learning as otherwise the agents are unable to explore the sparse competition reward." => One question that arises is the extent to which the outcome of learning is affected by this exploration reward and to explore its benefit. As already argued, we found it to be crucial for learning as otherwise the agents are unable to explore the sparse competition reward.

p8:
in a local minima => minimum

p9:
in references, you have Jakob Foerster and Jakob N Foerster => try to be more consistent.

p10, In Laetitia Matignon et al.  ... markov => Markov

p11, I would rename C_{alive} as C_{standing}

---

> ### Author Response · Authors · 2018-01-05
> **Response to Review**
>
> Thank you for taking the time to review our work. We have taken care of the typos and appropriate minor changes in the updated draft. We answer specific questions below.
>
> 1. “In Section 5.2, you are just describing "cool" behaviors observed from your videos ..... Study how the are represented in the networks? Anything beyond "look, that's great!" would make the paper better…”
> Reply:
> There are four main contributions put forth in this paper: (1) Identifying that a competitive multi-agent interaction can serve as a proxy for complexity of the environment and allows for a natural curriculum for training RL agents (2) Developing 4 new simulated environments which can serve as a test-bed for future research on training competitive agents (Section 3) (3) Developing opponent-sampling strategies and a simple (yet effective) strategy for dealing with sparse rewards (Section 4) -- both of which lead to effective training in the competitive environments (as evaluated through quantitative ablation studies in section 5.3 and 5.4) (4) Demonstrating compelling results on the four competitive 3D environments with two different agent morphologies -- showing remarkable dexterity in the agents without explicit supervision.
> We believe qualitative evaluation through observing agents’ behavior is an important part of evaluating the success of the proposed contributions, and in section 5.2 we analyze qualitatively many random episodes. Section 4 discusses the parallel implementation of PPO (a recent policy optimization algorithm, see section 2), opponent sampling and exploration curriculum which are crucial technical ideas making the results possible. Section 5.3, 5.4, 5.5 contain rigorous quantitative analysis of the main ideas which make these results possible.
> The results might be what one may expect, but executing on the idea is very much non-trivial -- as also noted by other reviewers. This paper is an exposé of the potential of competitive multi-agent training and we agree there is a lot of potential for more future work in this area.
>
> 2. “By the end of Section 5.2, you allude to transfer learning phenomena. It would be nice to study these transfer effects in your results with a quantitative methodology.”
> Reply:
> We studied a particular case of transfer in the sumo environment, where an agent trained in a competitive setting demonstrated robust standing behavior in a single-agent setting without any modifications or fine-tuning of the policies. The agent trained in a non competitive setting falls over immediately (with episodes lasting less than 10 steps on average), whereas the competitive agent is able to withstand strong forces for hundreds of steps. This difference (in terms of length of episode till agent falls over) can be quantified easily over many random episodes, we only included the qualitative results as the difference is huge and easily visible in the videos.
>
> 3. “In all subfigures in Figure 3, the performance of opponents should be symmetric around 50%. This is not the case for subfigures (b) and (c-1). Why? Do they correspond to non-zero sum game?”
> Reply:
> The games are not zero-sum, there is some chance of a draw as described in Section 3.
>
> 4. “Why do the last 2 images share the same caption?”
> Reply:
> Because the kick-and-defend game is asymmetric, so there are two plots -- one where keeper is trained with curriculum and another where kicker is trained with curriculum.
>
> 5. “I had a hard time understanding the message from Table 1. It really needs a line before the last row and a more explicative caption.”
> Reply:
> Added. It is also described in detail in Section 5.4
>
> 6. “Also, I find that annealing a kind of reward with respect to another is a weak form of curriculum learning. This should be further discussed.”
> This is discussed in section 4.1
>
> 7. “Actually, in Fig5: the axes are not labelled. I don't believe it shows a win-rate. So probably the caption (or the image) is wrong.”
> Reply:
> The y-label is the fractional win-rate (and not %), we have clarified this.
>
> 8. “would train on the dense reward for about 10-15% of the training epochs. So how much is \alpha_t? How did you tune it? Was it hard?”
> Reply:
> Note that \alpha_t is an annealing factor, so it’s value starts from 1 and is annealed to 0 over some number of epochs. 10-15% of training epochs is the typical window for annealing \alpha_t and the exact values are given in the experiments section 5.1. There is no direct tuning of \alpha_t required, instead the horizon for annealing was tuned in {250, 500, 750, 1000} epochs and the value giving highest win-rate was selected. More quantitative analysis of annealing is in section 5.3

---

### Official Review · AnonReviewer3 · 2017-11-26
**Impressive results; insights on opponent sampling**

**Rating:** 9
**Confidence:** 5

**Review:**

Understanding how-and-why complex motion skills emerge is an complex and interesting problem.
The method and results of this paper demonstrate some good progress on this problem, and focus on
the key point that competition introduces a natural learning curriculum.
Multi-agent competitive learning has seen some previous work in setting involving physics-based skills
or actual robots. However, the results in this paper are compelling in taking this another good step forward.
Overall the paper is clearly written and I believe that it will have impact.

list of pros & cons
+ informative and unique experiments that demonstrate emergent complexity coming from the natural curriculum
  provided by competitive play, for physics-based settings
+ likely to be of broad interest
- likely large compute resources needed to replicate or build on the results
- paper is not anonymous to this reviewer, given the advance publicity for this work when it was released
==> overall this paper will have impact and advances the state of the art, particular wrt to curriculums
    In many ways, it is what one might expect. But executing on the idea is very much non-trivial.

other comments

Can you comment on the difficulty of designing the "games" themselves?
It is often difficult to decide apriori when a game is balanced; game designers of any kind
spend significant time on this. Perhaps it is easier for some of the types of games investigated in
this paper, but if you did have any issues with games becoming unbalanced, that would be worthwhile commenting on.
Game design is also the next level of learning in many ways.  :-)

The opponent sampling strategy is one of the key results of the paper.
It could be brought to the fore earlier, i.e., in the abstract.

How much do the exploration rewards matter?
If two classes of agents are bootstrapped with different flavours of exploration rewards, how much would it matter?

It would be generally interesting to describe when during the learning various "strategies" emerged,
and in what order.

Adding sensory delays might enable richer decoy strategies.

The paper could comment on the further additional complexity that might result from situations
that allow for collaboration as well as competition. (ok, I now see that this is mentioned in the conclusions)

The Robocup tournaments for robot soccer (real and simulated) have for a long time provided
a path to growing skills and complexity, although under different constraints, and perhaps less interesting
in terms of one-on-one movement skills.

Section 2, "Notation"
why are the actions described as being discrete here, when the paper uses continuous actions?
Also, "$\pi_{\theta}$ can be Gaussian":   better to say that it *is* Gaussian in this paper.

"lead to different algorithm*s*"

Are there torque limits, and if so, what are they?

sec 4: "We do multiple rollouts for each agent *pair*" (?)

"Such rewards have been previously researched for simple tasks like walking forward and standing up"
Given the rather low visual quality and overly-powerful humanoids of the many of the published "solutions",
perhaps "simple" is the wrong qualifer.

Figure 2:  curve legend?

"exiting work" (sic)

4.2 Opponent sampling:
"simultaneously training"  should add "in opponent pairs" (?)

5.1 "We use both MLP and LSTM"
should be "We compare MLP and LSTM ..." (?)

For "kick and defend" and "you shall not pass", are there separate attack and defend policies?
It seems that these are unique in that the goals are not symmetric, whereas for the other tasks they are.
Would be worthwhile to comment on this aspect.

episodic length T, eqn (1)
It's not clear at this point in the paper if T is constant or not.

Observations: "we also give the centre-of-mass based inertia *tensor*" (?)

"distance from the edge of the ring"
How is this defined?

"none of the agents observe the complete global state"
Does this really make much of a difference?  Most of the state seems visible.

"But these movement strategies" -> "These movement strategies ..."

sec 5.4  suggest to use $\mathrm{Uniform}(...)$

"looses by default" (sic)

---

> ### Author Response · Authors · 2018-01-05
> **Response to Review**
>
> We thank the reviewer for carefully reading the paper and their encouraging feedback. We have taken care of the appropriate minor changes in the updated draft. We answer specific questions below.
>
> `1. "Can you comment on the difficulty of designing the "games" themselves?"
> Reply:
> We picked a few simple competitive games for these results. It is important to appropriately set termination conditions for the games. We did observe some difficulty in games becoming unbalanced, for example in kick-and-defend it was important to use a longer horizon for annealing exploration reward as the defender takes a longer time to learn and adding termination penalties for moving beyond the goal area as well as additional rewards for defending and not falling over on termination lead to more balanced win-rates. In future we will explore more complex game designs where the agents can make use of environmental resources to compete or have to overcome environmental obstacles in addition to competing.
>
> 2. "How much do the exploration rewards matter?"
> Reply:
> We analyzed the extreme cases where agents have or do not have an exploration reward, as well as the case when the exploration reward is never annealed. The summary is to use exploration reward but anneal it. We also experimented with additional reward terms for interaction with opponent in Sumo environment initially but didn’t observe any significant benefits and chose the simplest form of exploration rewards.
>
> 3. "Are there torque limits, and if so, what are they?"
> Reply:
> We used the default limits in the gym environment for ant and humanoid body, which is bounded between [-0.4, 0.4]
>
> 4. "For "kick and defend" and "you shall not pass", are there separate attack and defend policies? It seems that these are unique in that the goals are not symmetric, whereas for the other tasks they are."
> Reply:
> Yes, that is correct, we have noted this is section 5.1
>
> 5. "distance from the edge of the ring. How is this defined?"
> Reply:
> It is the radial distance of the agent from the edge. So if R is ring radius and r is the agent's distance from center then we give (R-r) as input.

---

### Official Review · AnonReviewer1 · 2017-11-28
**Review for Emergent Complexity via Multi-Agent Competition**

**Rating:** 7
**Confidence:** 4

**Review:**

This paper demonstrates that a competitive multi-agent environment trained with self-play can produce behaviors that are far more complex than the environment itself and such environments come with a natural curriculum by introducing several multi-agent tasks with competing goals in a 3D world with simulated physics. It utilizes a decentralized training approach and use distributed implementation of PPO for very large scale multiagent training. This paper addresses the challenges in applying distributed PPO to train multiple competitive agents, including the problem of exploration with sparse reward by using full roll-outs and use the dense exploration reward which is gradually annealed to zero in favor of the sparse competition reward. It makes training more stable by selecting random old parameters for the opponent.

Although the technical contributions seem to be not quite significant, this paper is well written and introduces a few new domains which are useful for studying problems in multiagent reinforcement learning. The paper also makes it clear regarding the connections and distinctions to many existing work.

Minor issues:

E[Loss] in table 1 is undefined.

In the notation section, the observation model is missing, and the policy is restricted to be reactive.

Uniform (v, \deta v) -> Uniform (\deta v, v)

---

> ### Author Response · Authors · 2018-01-05
> **Thank You**
>
> We thank the reviewer for carefully reading the paper and their positive feedback. We have taken care of the appropriate minor changes in the updated draft.

---

### Comment · AnonReviewer2 · 2018-01-06
**Lack of evaluation rigor**

There is a strong disagreement among reviewers (top paper according to one, clear rejection according to me), so I believe some discussion is necessary.

I completely agree that the videos are awesome, but in itself a nice video does not make a strong scientific paper. Engineering (and most of time a good deal of science) is about producing new interesting phenomena, but science is also about rigorously analyzing these results (rigorously generally means in a quantitative way) and extracting scientific messages from these analyses.

Quoting this excellent blog post
http://www.inference.vc/my-thoughts-on-alchemy/
"It's Okay to use non-rigorous methods, but... it's not okay to use non-rigorous evaluation".

Let me just take 3 examples to show you that this paper lacks rigor everywhere (the first two are admittedly minor points, but they feed my general feeling about this paper. Most other points are present in my review below, and the authors did not make the effort to address all of them):

Taken from their reply to my review:

1) "“In all subfigures in Figure 3, the performance of opponents should be symmetric around 50%. This is not the case for subfigures (b) and (c-1). Why? Do they correspond to non-zero sum game?”
Reply:
The games are not zero-sum, there is some chance of a draw as described in Section 3."

Well, in case of a draw, both players should get 0, so this cannot be a correct explanation of why the game is not zero sum. If I cannot trust the answer, can I trust the results themselves?

2) 4. "“Why do the last 2 images share the same caption?”
Reply:
Because the kick-and-defend game is asymmetric, so there are two plots -- one where keeper is trained with curriculum and another where kicker is trained with curriculum. "

Fine. I checked in the new version. The only sentence about Fig3 is "We plot the average win-rates over 800 games at various intervals during training in Fig. 3, for the sumo and kick-and-defend environments." Not a word about the above fact. The reader has to guess that. By the way, why don't we get any result for the "run to goal" case? What are the "various intervals"? Etc.

3) This one is much more serious, about transfer learning. Any rigorous transfer learning paper will precisely define a source domain and a target domain and will measure (quantitatively) the performance in the target domain with and without training first in the source domain. Here we just get "The agent trained in a non competitive setting falls over immediately (with episodes lasting less than 10 steps on average), whereas the competitive agent is able to withstand strong forces for hundreds of steps. This difference (in terms of length of episode till agent falls over) can be quantified easily over many random episodes, we only included the qualitative results as the difference is huge and easily visible in the videos."
OK, this is awesome. But to me, the evaluation methodology cannot just be "visual". This would be perfectly okay for the "scientific american" magazine, but this is not OK for a strong scientific conference. The evaluation itself has to be performed by the computer, because this makes it necessary to rigorously define all the conditions of this evaluation (define fall and stand, define the initial posture, define the number of steps you evaluate, define the wind variations, define everything). The computer will also be able to perform a statistical analysis, which is completely lacking here: did this phenomenon appear every time? If not, what are the chances? For instance, given the lack of quantitative analysis, one cannot determine if another method to come will perform better or worse.

To me, this paper is representative of what Ali Rahimi's talk was about (the talk was given after I wrote my review). So sorry guys, but though I like the videos and I must admit that my review is a little too harsh because the paper irritated me, I'm still considering that this paper should be rejected.

---

> ### Author Response · Authors · 2018-01-17
> **Response to further questions by Reviewer**
>
> We respond to the three points about the paper raised by the reviewer:
>
> 1) There are two questions here, one about games being zero sum and another about the plots in Figure 3 being symmetric about 50%. For the first question the answer is games are not zero sum and a draw results in a “negative” reward for both agents not 0 (considering it as a loss for both) and these rewards are very clearly defined in section 3 where the environments are introduced. Now, for the question about plots being symmetric, let's consider a simple example to understand why plot need not be symmetric. Say we are taking averages over 10 games, in the first set agent 1 wins 5 and 5 are draws (considered as losses for both), in the second set agent 1 wins 6 and and rest are draws. Then a curve representing average win-rates of agents over these sets of games will have two points for agent 1 at 50% and 60%, thus giving an increasing curve for agent 1 whereas for agent 2 the curve will be 0. It is easily seen in this synthetic example that the curve for two agents is not symmetric.
>
> 2) The legend for the plots clarifies this, where the curves are labelled as kicker:no-curriculum, keeper: curriculum and kicker: curriculum, keeper:no-curriculum respectively for the two plots. Since only one of the reviewer seemed to have a problem understanding this plot we didn't add any additional clarification. We will add a line clarifying this further.
>
> 3) Note that we have not highlighted this as a contribution of the paper anywhere in the abstract or introduction. This is also in section 5.2 which we have explained earlier is the qualitative evaluation part of the paper. The evaluation of transfer success (falling or not) is performed by computer code that does returns objective result of the episode. We do have quantitative evaluation for experiment reviewer asks, but do not report the numbers because the difference between using our approach and not is very stark. If other reviewers agree, we are happy to include these numbers in the final manuscript.

---

### Decision · Program_Chairs · 2018-01-29
**ICLR 2018 Conference Acceptance Decision**

**Decision:**

Accept (Poster)

**Comment:**

This paper received divergent reviews (7, 3, 9). The main contributions of the paper -- that multi-agent competition serves as a natural curriculum, opponent sampling strategies, and the characterization of emergent complex strategies -- are certainly of broad interest (although the first is essentially the same observation as AlphaZero, but the different environment makes this of broader interest).

In the discussion between R2 and the authors, I am sympathetic to (a subset of) both viewpoints.

To be fair to the authors, discovery (in this case, characterization of emergent behavior) can be often difficult to quantify. R2's initial review was unnecessary harsh and combative. The points presented by R2 as evidence of poor evaluation have clear answers by the authors. It would have been better to provide suggestions for what the authors could try, rather than raise philosophical objections that the authors cannot experimentally rebut.

On the other hand, I am disappointed that the authors were asked a reasonable, specific, quantifiable request by R2 --
"By the end of Section 5.2, you allude to transfer learning phenomena. It would be nice to study these transfer effects in your results with a quantitative methodology.”
-- and they chose to respond with informal and qualitative assessments. It doesn't matter if the results are obvious visually, why not provide quantitative evaluation when it is specifically asked?

Overall, we recommend this paper for acceptance, and ask the authors to incorporate feedback from R2.